# The Development of Coal Mine Methane Utilization Infrastructure within the Framework of the Concept "Coal-Energy-Information"

**Arina Smirnova** [1,2]**, Kirill Varnavskiy** [1]**, Fedor Nepsha** [1,]*****, Roman Kostomarov** [1] **and Shaojie Chen** [2]

[1] Mining Industry Digital Transformation Lab, Mining Institute, T.F. Gorbachev Kuzbass State Technical University, 650000 Kemerovo, Russia

[2] State Key Laboratory of Mining Disaster Prevention and Control Co-Founded by Shandong Province and The Ministry of Science and Technology, Shandong University of Science and Technology, Qingdao 266590, China

***** Correspondence: nepshafs@kuzstu.ru

**Abstract:** The operation of coal mines is intricately linked with emitting a large quantity of coal mine methane, and in most cases, this methane releases into the atmosphere. In total, according to statistics, coal mining enterprises emit 8% of anthropogenic methane, determining a contribution to greenhouse gas emissions to the amount of 17%. There are various means for coal mine methane utilization. In this study, the concept "Coal-Energy-Information" is proposed. This concept implies both the construction of data processing centers on the industrial sites of coal mines and the usage of coal mine methane. Coal mine methane can be used as a primary energy source for the energy supply of data processing center consumers as well as coal mine consumers with necessary energy resources (electricity, heat, and cooling). Within the framework of the proposed concept, three options of coal mine methane utilization are considered. The first option is the use of gas genset for electrical and thermal energy generation (cogeneration) and their usage for coal mine and constructed data processing centers and consumers' power supply. The second option is absorption refrigerator usage (with coal mine methane direct burning) for cooling the IT equipment of constructed data processing centers. The last one is the use of a gas genset and absorption refrigerator (trigeneration) for constructed data processing centers' and coal mine consumers' energy supplies (electricity, heat, and cooling). In conclusion, it is noted that proposed concept is closely correlated with the program for the development of the coal industry in Russia for the period up to 2035, since it allows creating a base for the implementation of innovative technologies based on digital platforms that ensure the development of coal mining technology without the constant presence of personnel in underground mining facilities.

**Keywords:** coal mines; data processing centers; trigeneration; coal mine methane utilization

## 1. Introduction

According to the results and reports of 2021, the coal production in Russia almost reached the level of 2019 and was more than 435 million tons, virtually 10% higher than that in 2020. Therefore, the dynamics of the huge surge in coal demand in the world market has been noticed in the extraction volume and national coal costs over the period of 2021.

Figure 1 provides current data about the coal costs per ton of three global futures [1]. According to the given information, the behavior of steam coal costs for the period from 2021 to present was ambiguous. It is shown that in the second half of the year, coal costs remained approximately stable and were USD 220 per ton. By the end of 2021, the Rotterdam and Newcastle Coal Futures prices began to increase rapidly. The reason for this was the abolishing of anti-COVID restrictions in the world and the growth in demand for primary energy sources. However, containment measures are still being taken occasionally

in China. Due to that fact, the significant decline in production in China is reflected in the steady cost of coal.

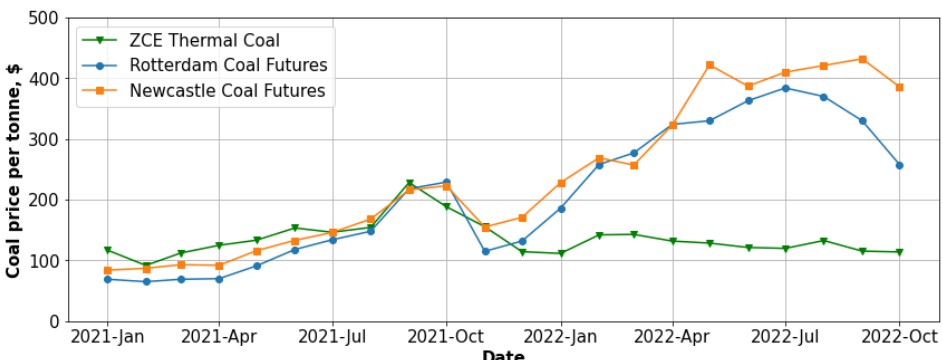

**Figure 1.** Trends and forecasts of world coal prices [1].

There was a spike in coal costs because of aggravation of the international energy crisis in the first half of 2022. Therefore, the peak of Newcastle Coal Futures prices was reached in September and was USD 420 per ton, and for Rotterdam Coal Futures, its peak was in July with a USD 390 per ton price.

The ambiguous dynamics of coal prices have a negative impact on the economic stability of coal mining enterprises. In this regard, an urgent task arises: finding alternative ways to ensure economic sustainability using the existing infrastructure of coal mining enterprises.

According to most forecasts, the consumption of coal will not grow significantly anymore. Moreover, it will slow down in the future, and it seems logical to use the current period of high prices for coal to search for new techniques for business diversification of coal mining enterprises, because the obtained windfall can be directed to the creation and implementation of new technologies.

The most obvious way to introduce technologies for coal deep processing is to use the industrial site of coal mining enterprises whenever possible. However, this method requires significant capital investments, while the markets for such products are not always clear.

Another promising option is the implementation of various technologies within the "Coal-Gas-Electricity" concept [2], with the production of electrical energy at the output.

This option certainly has great prospects for practical implementation, but according to the authors, it can be supplemented and expanded, namely by the creation of technologies within the framework of the "Coal-Energy-Information" concept. In practice, this implies the use of infrastructure and production facilities of coal mining enterprises for the construction and provision of the engineering infrastructure of data processing centers (DPCs) with the necessary energy resources.

Currently, there is constant growth in the IT industry. The volume of processed data is increasing, and consequently, the volume of electricity consumption of DPCs is also growing [3]. Therefore, the growing demand for DPCs leads to a construction cost increase.

Energy consumption by DPCs is significant, and it can be said that DPCs belong to the energy-intense industrial energy consumers. The share of electrical energy costs is about 40% of the total OPEX. Taking into consideration the trend to use clean energy sources, in [4], the need to use renewable energy sources to create "green" DPCs is mentioned. However, the use of "traditional" renewable energy sources in Kuzbass (the main coal-producing region in Russia) is far from constant cost effectiveness.

However, for Kuzbass, it is relevant to use one the most ecologically clean sources from the hydrocarbon ones: coal bed methane (CBM). The coal deposits in Kuzbass contain large amounts of CBM comprising methane from 80% to 95%, with lower contents of such heavier hydrocarbons as ethane and propane and nonhydrocarbon gases such as nitrogen and carbon dioxide [5].

The Kuznetsk Basin, situated in the southern part of Western Siberia, is the most mature basin for CBM exploration and development in Russia, with prospective original in-place hydrocarbon reserves of 13.1 trillion $m^3$ [6]. Thus, CBM can be simultaneously not only a hazardous industrial factor in coal mining but also a valuable co-product for recovery and utilization [7].

CMM is a type of CBM released as a direct result of the physical process during underground mining works and coal seam extraction. According to the authors of [8], coal mines emit 8% of their anthropogenic methane emissions during extraction and processing activities. In the first half of 2022, Kuzbass mining enterprises extracted 38 million tons with underground methods, while the average methane emissions per 1 ton of extracted coal was 17 $m^3$/t [9]. The rapid development of the coal mining industry led to the increase in mining depth, which remarkably affects environmental pollution with methane emissions. Because of the high CMM quantities emitted into the environment, methane utilization is an advanced and relevant direction for coal region development. For greenhouse gas emissions mitigation, methane could be captured and used for power generation [10].

Therefore, the use of one more alternative sources of energy (CMM) for power generation is of growing interest. According to the authors of [11], the cost of power generation using CMM is 30–50 percent lower than the cost of power generation produced by wind power plants. At the same time, the investments to eliminate 1 ton of equivalent annual carbon dioxide emissions when using CMM are USD 34, which is four times lower than the CAPEX of wind farms (USD 100–142).

In general, three main factors determine the feasibility of CMM utilization: (1) reduction of the number of MAM explosions during underground coal deposit development and improving the safety of mining activities; (2) new job formation at gas fields and gas processing enterprises and improving the economic performance of coal mines; (3) improving the environmental situation in coal mining regions by the reduction of greenhouse gas emissions, one of which is methane. (The global warming potential of methane is 21 times greater than that of carbon dioxide.)

There are many studies devoted to the issue of CMM processing [8,12–19]. In general, they are aimed at ensuring the maximum use of CMM released from coal seams with efficiency of up to 95%. For this purpose, a gas genset operating in a trigeneration cycle can be used. The trigeneration cycle allows providing an electrical power supply to a coal mine and DPC, as well as providing a heat supply to a coal mine and cooling to the IT equipment of a DPC. In the future, the deep processing of CMM can provide production of carbon dioxide, which can be used in agriculture.

Within the framework of this article, the authors considered three options for the utilization of CMM:

1. The use of a gas genset for generation of electrical and thermal energy (cogeneration);
2. The use of a gas genset and absorption refrigerator (AR) for an energy supply to a DPC constructed on the industrial site of a coal mine (trigeneration);
3. The use of an AR (with direct burning of CMM) for the cooling of IT equipment of constructed DPCs.

This article is structured as follows. In Sections 2–4, options for the utilization of CMM are considered. Sections 5 and 6 provide a comparison of three methods of CMM utilization and considers the stakeholders interested in the development of the CMM utilization sector and the deployment of DPCs. Section 7 provides the conclusion of the article.

## 2. Energy Supply System Based on a CMM-Fired Gas Genset

### 2.1. Common Information

In contrast to traditional gasfields, CBM is an unconventional methane resource, and methane is not in a free form in the porous medium but in a bound (sorbed) form, which is stored in the natural fractures (cleats) and coal micropores [20,21]. The coal seam methane content is the volume of methane contained in a unit volume of coal (in $m^3$/t) on a dry ash-free (daf) basis. The values for the coal seam methane content in the Kuznetsk Basin

area reach more than 30 m$^3$/t on a daf basis and rise with increasing coal deposit depths and decreasing heat and humidity [16]. Currently, the normative and technical documentation obliges Russian users of subsurface resources to carry out coal seam degassing work when the natural gas content exceeds 13 m$^3$/t on a daf basis. Therefore, CMM utilization is becoming required because of this rule.

The volume of methane released during mining operations in a unit of time is characterized by the methane-bearing capacity in m$^3$/min. According to the authors of [22], the yearly average relative methane-bearing capacity in Kuzbass coal mines is from 11 to 150 m$^3$/min while mining coal at a rate of 1600–13,074 t/day. The values of the relative methane-bearing capacity of the most dangerous mines in Kuzbass are shown in Figure 2 in descending order [23].

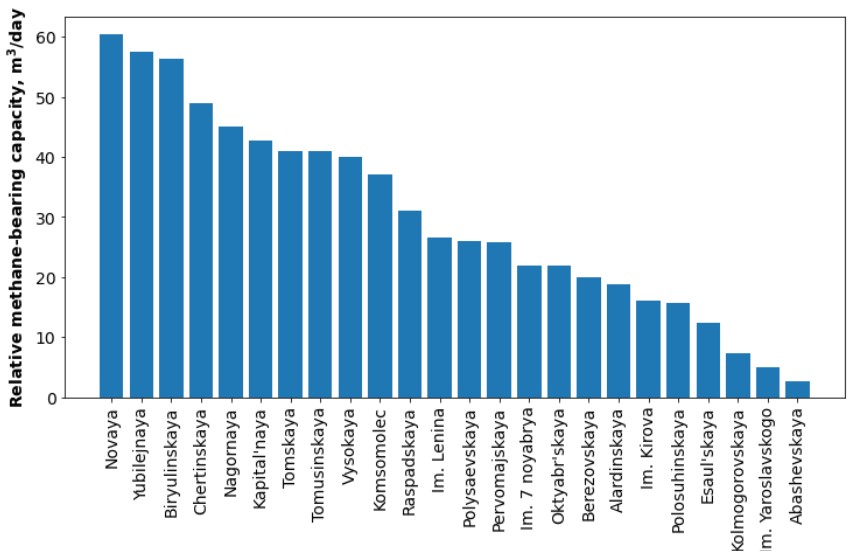

**Figure 2.** Relative methane-bearing capacity of the Kuzbass coal mines [23].

### 2.2. Analysis of the Composition of GAM

Figure 3 presents a monthly graph of the amount of a methane-air mixture removed from underground mine operations, as well as the concentration of methane from one of the coal mines of Kuzbass. The case of this coal mine is considered in this paper.

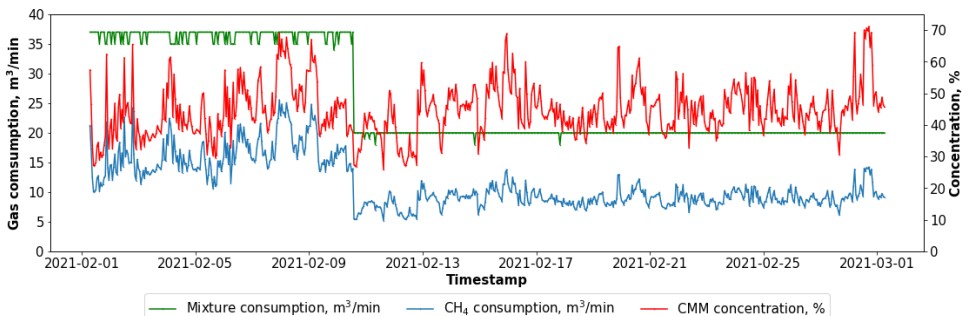

**Figure 3.** Monthly graph of changes in methane concentration and volumes of GAM removed from the mine.

These graphs show that the methane concentration did not drop below 30%, which indicates that there is no need to enrich GAM for use as a fuel for a gas genset. At the same time, it is necessary to note the sharply variable nature of the change in concentration, which was caused by a change in the concentration of methane in the process of coal mining.

A structural diagram of a power system with a gas genset is presented in Figure 4.

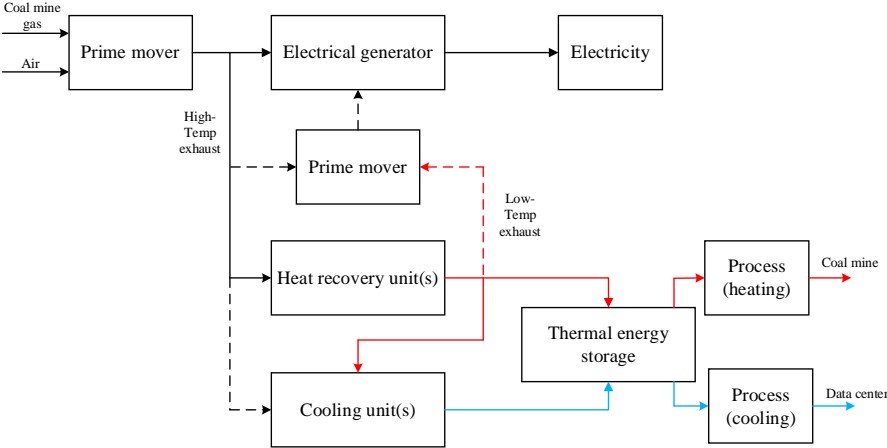

**Figure 4.** Block diagram of CMM utilization using gas genset.

The diagram in Figure 4 allows one to determine the approach to the choice of GPU: (1) analysis of the composition of the gas-air mixture (GAM); (2) assessment of the calorific capacity of the gas mixture; (3) assessment of amount of generated electrical and thermal energy; and (4) the choice of gas genset.

### 2.3. Algorithm of Choosing the CMM Utilization Scheme

This paper assumes the use of a mine infrastructure for the construction of DPCs, which will be the main consumer of energy resources produced using CMM utilization. Figure 5 provides the algorithm for choosing the optimal CMM utilization method with this approach.

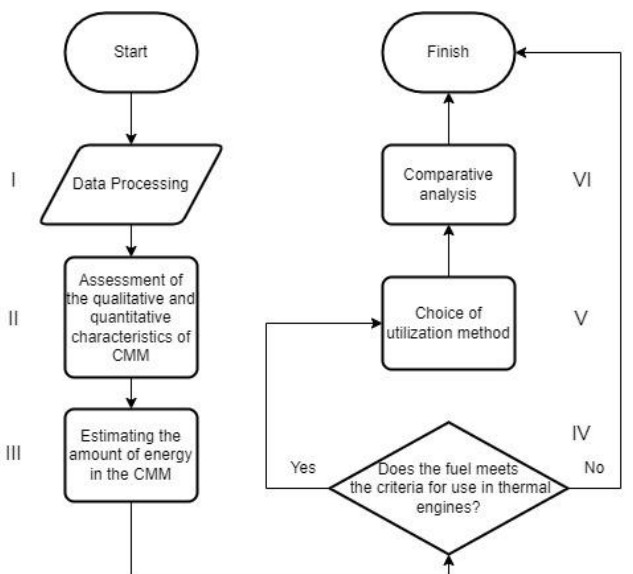

**Figure 5.** Algorithm for choosing a method of utilization of CMM.

The algorithm in Figure 5 includes the following stages:

*I*. Obtaining data on the volume and composition of the gas-air mixture and the CMM concentration.

*II*. Analysis the composition of the gas-air mixture for the needed filtration and dehumidification. In general, the CMM must meet the main criteria specified in Table 1. If the permissible values are exceeded, then filtering and dehumidification equipment must be used.

**Table 1.** The main criteria for the qualitative composition of CMM. The data in the table are given in accordance with the state standards of the Russian Federation.

| Index | Unit | Admissible Value |
|---|---|---|
| Sulfur | $g/m^3$ | 0.036 |
| $H_2O$ | $mg/m^3$ | 9 |
| Solid impurities | $mg/m^3$ | 1 |
| Particle size of solids | micron | 10 |

*III*. At this stage, the calorific value of the captured gas-air mixture is calculated.

*IV*. Determine the possibility of using CMM as a fuel for thermal engines. The main evaluation criteria are the calorific value, the concentration of methane in the mixture (not less than 25%) and the volume of the captured gas-air mixture. Based on the above data, it is possible to determine the type and capacity of equipment for the utilization of CMM.

*V*. Next, the choice of utilization method for CMM is carried out. In this paper, three ways to utilize CMM are considered:

- DPC with cogeneration energy center (CEC): The CEC consists of a gas genset operating on a cogeneration cycle and providing the DPC and coal mine with electrical and thermal energy. In this case, excess heat can be given off to meet the mine's own needs.
- DPC with refrigeration from an AR: In this case, it is proposed to use a direct-combustion AR to provide cooling to the DCP.
- DPC with a trigeneration energy center (TEC): The TEC combines the advantages of a CEC and AR and provides the DPC with electricity, heat and cooling.
- At this stage, the selection of equipment and a feasibility study for each specific case are also carried out.

*VI*. Conduct a comparative analysis to determine the most effective method.

Each of the stages is discussed in more detail below.

### 2.4. Assessment of the Calorific Capacity of GAM

For determination of the gas genset power when operating on CMM, it is necessary to estimate the potential amount of energy contained in cubic meters in the GAM which ventilates from the mine. The drive engine of a gas genset is an internal combustion engine, and therefore, the analysis of the energy potential of GAM must be carried out based on the assessment of the calorific value of GAM and its debit [24].

To assess the calorific value of GAM, it is proposed to use Equation (1) of D.I. Mendeleev:

$$LCV_f = LCV_1 \cdot a_1 + LCV_2 \cdot a_2 + \ldots LCV_n \cdot a_n = \sum_{i=1}^{n} LCV_i \cdot a_i \tag{1}$$

where $LCV_f$ is the lower heating value of fuel ($MJ/m^3$), $LCV_i$ is the lower heating value of gases in the composition of GAM ($MJ/m^3$), and $a_i$ is the share of *i*-gas in GAM.

### 2.5. Assessment of the Amount of Generated Electrical and Thermal Energy

The lower heating value of GAM can be calculated by the following Equation (2):

$$LCV_v = 60 \cdot R \cdot LCV_f \tag{2}$$

where $R$ is the debit of GAM ($m^3$), $LCV_f$ is the lower heating value of fuel ($MJ/m^3$), and $LCV_v$ is the lower heating value of GAM ($MJ/m^3$).

From the heat balance in Equation (3) and approximate heat balance values (Table 2), it can be seen that no more than 28% of the fuel energy in a gas internal combustion engine can be transformed into effective power and then into electrical energy:

$$Q_0 = Q_e + Q_c + Q_{exh} + Q_{icf} + Q_{unc} \tag{3}$$

where $Q_e$ is the heat equivalent to efficient engine operation (MJ/h), $Q_c$ is the heat given off to the cooling medium (MJ/h), $Q_{exh}$ is the heat carried away from the engine with exhaust gases (MJ/h), $Q_{icf}$ is the heat lost due to incomplete combustion of fuel (MJ/h), and $Q_{unc}$ is the unaccounted heat loss (MJ/h).

**Table 2.** Approximate heat balance of internal combustion engines (%) [25].

| Engine | $Q_e$ | $Q_{exh}$ | $Q_c$ | $Q_{icf}$ | $Q_{unc}$ |
|--------|-------|-----------|-------|-----------|-----------|
| Petrol | 21–28 | 30–55 | 12–20 | 0–45 | 3–8 |
| Gas | 23–28 | 35–45 | 20–25 | 0–5 | 5–10 |
| Diesel | 29–45 | 25–45 | 15–35 | 0–5 | 2–5 |

*2.6. The Choice of Genset Type and Feasibility Study*

According to the obtained values of $LCV_e$, it is possible to select the type and composition of a gas genset.

For the feasibility study, Equations (4)–(9) were used.

The fuel costs in USD/kW·h are expressed as

$$C_{fuel} = \frac{R_{gas} \cdot C_{gas} \cdot K_{up}}{P_{unit}}, \tag{4}$$

where $R_{gas}$ is the gas consumption (m$^3$/h), $C_{gas}$ is the gas cost, (when using CMM, it is conventionally taken as equal to USD 0), $P_{unit}$ is the power of the gas genset (kW), and $K_{up}$ is the capacity factor.

The maintenance costs in USD/kW·h are expressed as

$$C_{oil} = \frac{V_{oil} \cdot C_{oil1}}{T_m \cdot P_{unit}}, \tag{5}$$

where $V_{oil}$ is the oil volume (l), $C_{oil1}$ is the cost of 1 L of oil (USD), and $T_m$ is the frequency of maintenance in running hours.

The service costs, including overhaul, in USD/kW·h are expressed as

$$C_m = \frac{C_{m1}}{T_{oh} \cdot P_{unit}}, \tag{6}$$

where $C_{m1}$ is the service costs, including overhaul (USD), and $T_{oh}$ is the frequency of overhaul in running hours.

The oil burning costs in USD/kW·h are expressed as

$$C_{ob} = \frac{R_{ob} \cdot C_{oil}}{P_{unit}}, \tag{7}$$

where $R_{ob}$ is the oil consumption for burning (gr/kWh).

The costs for spare parts, including overhaul, in USD/kW·h are expressed as

$$C_{sp} = \frac{C_{sp1}}{T_{oh} \cdot P_{unit}}, \tag{8}$$

where $C_{sp1}$ is the cost of consumables and spare parts, including overhaul (USD). ($C_{sp1}$ is accepted to be equal to 50% of the total cost of installation.)

Amortization in USD/kW·h is expressed as

$$C_{dep} = \frac{C_{unit}}{T_{oh} \cdot P_{unit}}, \tag{9}$$

where $C_{unit}$ is the full cost of the gas genset and auxiliaries' systems (USD).

Significantly increasing the efficiency of a gas genset is possible using an exhaust gas heat and coolant recovery system.

When calculating the thermal correction, it is accepted that there will be a replacement of an existing heat source in a gas heat and coolant recovery system. The capacity factor at this stage is neglected.

The thermal correction ($/kWh) is the cost of 1 kW of thermal energy produced from an existing source (boiler room). In the case of a cogeneration cycle, it is replaced by the waste heat of the exhaust gases of the gas genset. This can be determined according to Equation (10):

$$C_{heat} = \frac{R_{fuel} \cdot C_{fuel}}{Q_{unit}}, \tag{10}$$

where $C_{fuel}$ is the cost per m$^3$ (ton) of fuel for the current heat source (USD), $Q_{unit}$ is the thermal capacity of a heat source (kW), and $R_{fuel}$ is the fuel consumption.

### 2.7. Calculation of Greenhouse Gas Emissions

For calculation of the reduction of greenhouse gas emissions, it is necessary to normalize to the amount of emissions of greenhouse gases into units (global warming potential). The *GWP* of methane is 25 units.

Thus, reducing emissions from CMM utilization based on gas gensets can be calculated by Equation (11):

$$CO_{2reduced} = \frac{0.6682 \cdot R_{cmm} \cdot \%CH_4}{1000} \cdot GWP, \tag{11}$$

where 0.6682 is the methane density (kg/m$^3$), $R_{cmm}$ is the gas consumption of a gas genset (m$^3$/h), and %CH$_4$ is the percentage of a greenhouse gas (methane) in GAM.

In accordance with [26], the average cost of CO$_2$ emissions in 2021 was USD 58.97/ton, and the calculation of savings on quotas is produced by Equation (12):

$$ETS_{year} = CO_{2reduced} \cdot P_{ETS} \tag{12}$$

where $P_{ETS}$ is the cost of CO$_2$ emissions.

The cost calculation in USD/kWh is shown in Equation (13):

$$C_{kW} = C_{gas} + C_{oil} + C_m + C_{ob} + C_{sp} + C_{dep} - C_{heat}. \tag{13}$$

$C_\Delta$ in USD/kWh is calculated as shown in Equation (14):

$$C_\Delta = C_{kW} - Rate_e, \tag{14}$$

where $C_\Delta$ is the difference in the cost of 1 kW of energy produced by a cogeneration gas genset and the current electricity rate and $Rate_e$ is the electricity rate (USD/kWh).

The economic effect in USD per year is calculated as shown in Equation (15):

$$E = C_\Delta T_{eng} \cdot P_{unit} \cdot K_{up}, \tag{15}$$

where $T_{eng}$ is the running hours.

The payback period in years is calculated as shown in Equation (16):

$$S = \frac{C_{unit}}{E} \tag{16}$$

The calculation results are shown in Table 3. The calculations were made according to the average values of the data for the volume and concentration of methane presented in Figure 3.

**Table 3.** Calculation results.

| Parameter Name | Symbol | Unit | Value |
|---|---|---|---|
| Calorific capacity | $LCV_f$ | MJ/m$^3$ | 13.24 |
| Amount of energy | $LCV_v$ | MJ/m$^3$ | 30,200.88 |
| Effective power | $Q_e$ | MJ/h | 1812.05 |
| Maintenance costs | $C_{oil}$ | USD/kW·h | 0.0016 |
| Service costs, including overhaul | $C_m$ | USD/kW·h | 0.0055 |
| Oil burning costs | $C_{ob}$ | USD/kW·h | 0.00052 |
| Costs for spare part, including overhaul | $C_{sp}$ | USD | 329,676.34 |
| Amortization | $C_{dep}$ | USD/kW·h | 0.01 |
| The thermal correction | $C_{heat}$ | USD/kW·h | 0.01 |
| Savings on quotas | $ETS_{year}$ | USD | 70,302.49 |
| Cost price | $C_{kW}$ | USD/kW·h | 0.0076 |
| Economic effect | $E$ | USD/year | 360,798.50 |
| Payback period | $S$ | year | 5 |

*2.8. Effects of the Considered Option*

The use of a gas genset for the generation of electrical and thermal energy (cogeneration) allows achieving the following:

1. Reduction of payments for electricity consumed from the grid (by reducing the maximum power and reducing the consumption of electrical energy);
2. Operation of a gas genset in cogeneration mode allows providing heating and a hot water supply to industrial site consumers (replacement or alternative to a boiler room);
3. Reduction of fees for greenhouse gas emissions (methane);
4. Increasing a power supply's reliability.

**3. DPC Construction in Addition to the Gas Gensets**

To improve the reliability of the DPC and coal mine energy supply, CMM utilization can be used. In this case, the coal mine energy supply structure is as shown in Figure 6.

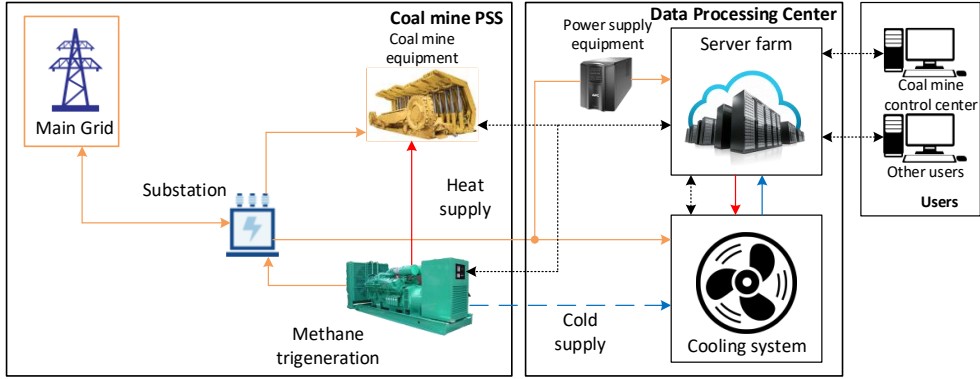

**Figure 6.** Structure of power supply system of coal mine and DPC.

In our case, it was assumed that the DPC equipment was located on the surface. However, it is possible to place the IT equipment in the underground part of the coal mine. One such example is the DPC placed in Norway's Lefdal mine, where olivine, a mineral used to make heat-resistant glass, was previously mined [27]. It must be noted that the DPC equipment can be used to build applied control systems for the technological systems of a coal mine [28].

The cooling system is the second consumer in the DPC after computer equipment. Therefore, it is quite useful to use gas piston units in trigeneration mode. In this case, they can provide refrigeration within a coal mine and significantly reduce the Power Usage

Efficiency (PUE), which is a ratio that describes how efficiently a computer DPC uses energy in comparison with cooling and other overhead.

The share of other consumers such as lighting, alarm, control, and monitoring systems depends on the DPC size and is about 6% of the total consumption of the DPC.

*3.1. Options for the Use of DPC by Enterprises of the Mineral Resources Sector*

According to the authors of [29], DPC leases are grouped into four categories:

1. *Wholesale or powered shell leases* are the provision of premises and infrastructure for the power supply of a DPC building. These leases are typically long-term contracts due to the significant investment made by the tenant.
2. *Turnkey or enterprise leases*, where the landlord builds the shell building and installs all infrastructure needed, including the raised floor. The tenant has a server room ready for operation at his disposition.
3. *Colocation agreements* consist of providing a fully equipped server room, including the server racks.
4. *Cloud agreements (IaaS)* offer IT services to a wide range of tenants. The tenant rents computing power and is not involved in the operation of DPC equipment.

The second and third options are preferable for coal mining enterprises. The DPC construction can be entrusted to an external, more experienced company when implementing the second option. During the transition to the third option, it is assumed that the enterprise will have personnel serving the life support system of the DPC. In the future, it is possible to switch to the fourth option, where an enterprise will be able to provide cloud services. These services can be related to the automation of business processes of other regional enterprises. This includes platform services (IaaS) and application components (SaaS).

*3.2. DPC Construction Cost Calculation*

The DPC construction cost calculation (CAPEX) is a rather complicated process. Therefore, for this purpose, a special service Data Center Capital Cost Calculator was used [30].

The initial parameters for the calculation are presented below:

1. *DPC environment*

Location of data center: Europe, Bulgaria. (The choice of this country was justified by the lack of Russia in the Data Center Capital Cost Calculator. Therefore, it was decided to choose the country with the closest value for the average gross monthly wage (Russia = USD 1046, Bulgaria = \$885 in 2022).

Data center design capacity: 1000 kW.
Cooling system: computer room air handler (CRAH).
UPS architecture: traditional, non-scalable UPS.
Power distribution type: basic wall-mount panelboards.
Power density: 5 kW.
Cost of work for one person hour: USD 22.5/h.
CAPEX: USD 3,000,000.

2. *Redundancy level*

DPC redundancy is typically described in four groups with increasing resilience to component failure: N, N + 1, 2N, and 2N + 1. The N set-up is configured to have just the number of components it needs to function, meaning that whenever one component fails, the entire system fails. The N + 1 configuration indicates that there is one extra component on site regardless of the size of N.

The price includes the costs of the racks, raised floor, fire suppression, switchgear, and dropped ceiling.

As a result, the projected data center included 200 racks and was located on an area of 664 m$^2$, while the building area was 1135 m$^2$. At the same time, the construction cost was EUR 3 million.

According to the authors of [29], the operating costs were USD 225/m$^2$ per month.

### 3.3. Determination of DPC Profitability

Colocation agreements are considered when determining the yield. Here, 5 kW racks were leased, and the Tier 3 reliability level was provided. According to the authors of [29], the rental price was USD 800 per month. Therefore, it was assumed that 70% of the 200 racks were always leased. In this case, the annual OPEX will be USD 255,375, and the rental income from the racks was USD 1,920,000. The payback period at a discount rate of 12.5% (for the Russian Federation) is about 4 years.

### 3.4. Effects of the Considered Option

In addition to the previously considered option, the following effects will be obtained:

1.  Business diversification toward the IT sector with the possibility of generating additional income;
2.  The ability to provide cooling to DPC IT equipment when using the gas-generating set in trigeneration mode;
3.  The ability to achieve the maximum efficiency of the generating unit up to 85%;
4.  Development of the regional IT industry by building a new DPC to use local deposits and employment creation.

## 4. DPC Construction Together with Absorption Refrigerator

IT equipment generates a lot of heat during operation. Therefore, the cooling of the DPC premises, in which the IT equipment is located, is the most important task to ensure the reliability of the DPC operation. In turn, energy consumption for the cold supply is, on average, 40–45% of the total energy consumption.

The use of AR in the cooling system of a DPC located on the territory of a coal mine will significantly reduce energy costs for cooling IT equipment. In this case, it is proposed to use AR with direct combustion while using CMM as a primary fuel.

The power supply structure of a coal mine and DPC when using an AR is shown in Figure 7.

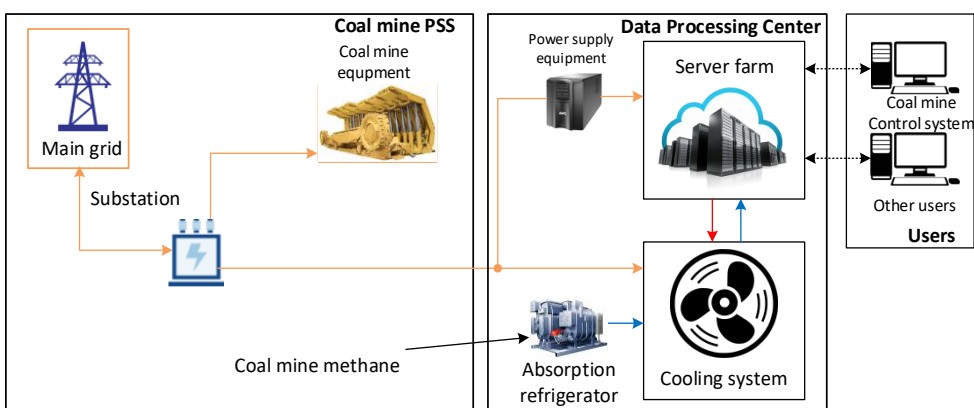

**Figure 7.** Power supply structure of coal mine and DPC when using ARs.

With direct combustion of the primary fuel, it is most efficient to use two-stage ARs. The cooling capacity of an AR can be determined by Equation (17):

$$Q_{cool} = Q_{heat} \cdot COP, \tag{17}$$

where *COP* is the refrigerating factor (for a two-stage AR, COP can be taken to be equal to 1.2).

The heat from the burner to the AR can be calculated by Equation (18) [13]:

$$Q_{bur} = LCV_{CMM} \cdot Rate_{CMM}, \tag{18}$$

On average, according to the catalog, the consumption of AR (with direct combustion) with a refrigerating capacity of 1.1 MW was approximately 85 m$^3$/h. However, the use of natural gas was assumed. Preservation of the thermal power of the AR burner during the combustion of CMM implies the introduction of appropriate changes in the burner design, as well as an increase in gas consumption (the amount of gas consumed depends on its composition) [24].

Using Equation (18), the amount of heat supplied from the burner was determined when using natural heat for the considered AR $Q_{bur} = 778$ kW. When using CMM with a net calorific value of 18.38 MJ/m$^3$, a supply of 152.64 m$^3$/h of coal mine methane (or 2.53 m$^3$/min) would be required to operate a similar AR.

The amount of methane captured from the mine in question was four times higher than that required for the operation of the selected AR and, accordingly, the cooling supply of the DPC.

During the cold season, when the cooling of IT equipment can be carried out using free cooling, an AR can be used for the heating and hot water supply of the administrative premises of the DPC and mine, but the efficiency of the AR in heating mode will decrease.

*Effects of the Considered Option*

1.　Business diversification toward the IT sector, with the possibility of generating additional income;
2.　The possibility to provide effective CMM utilization year-round;
3.　Using AR for the heating and hot water supply of industrial site consumers (replacement for or alternative to a boiler room);
4.　Reduction of fees for greenhouse gas emissions (methane).

## 5. Common Results

The CAPEX of DPC construction was USD 3,000,000. The OPEX was USD 255,000 per year. With a gas genset (cogeneration), AR, or gas genset together with an AR (trigeneration) installation, it became possible to decrease operational expenditures. Table 4 presents the comparison of CMM utilization options.

**Table 4.** Comparison of CMM utilization options.

| Parameter Name | DPC with Cogeneration Energy Center (CEC) | AR + DPC | DPC with Trigeneration Energy Center (TEC) |
|---|---|---|---|
| Additional CAPEX (USD) | 672,000 | 345,000 | 1,140,000 |
| CAPEX decrease (USD) | - | 200,000 | 200,000 |
| Additional OPEX (USD) | 67,000 | 10,000 | 77,000 |
| Profitability or Savings (USD) | 200,000 | 60,000 | 260,000 |
| Total OPEX (USD) | 122,000 | 205,000 | 72,000 |
| Payback period of CMM utilization equipment | ≈5 years | ≈7 years | ≈6 years |

Table 4 shows that all proposed CMM utilization options allowed decreasing the total OPEX by lowering electricity and heating bills or providing cooling for the computer room. The AR and trigeneration energy center replaced some parts of the cooling system, such as the chiller and cooling tower. Therefore, the CAPEX was lowered by USD 200,000. The choice of CMM utilization option depends on the geological conditions of the coal mine and on electricity and heat tariffs. For the considered use case, the third option yielded the best result.

## 6. Discussion

The following benefits can be achieved through implementation of proposed CMM utilization options:

1. Improving labor safety at underground coal mines.
2. The possibility of using industrial sites of closed (or mothballed) underground coal mines and partial preservation of jobs.
3. An increase in tax revenues for the regional budget from organizations that own DPCs, as well as service companies that maintain and operate DPCs.
4. The creation of additional job opportunities in one of the most advanced industries: the IT sector.
5. Improving the environmental situation by reducing greenhouse gas emissions.
6. Development of the regional IT sector through the creation of infrastructure facilities, namely DPCs with increased energy efficiency in the local resource base.
7. The possibility of creating environmentally friendly heat supply facilities for the population (with the usage of heat removed from server farms of DPCs).
8. Improving the efficiency of the development of coal deposits through an integrated approach (the use of coal and CMM) and increasing the profitability of coal mining.
9. Diversification of the business of coal companies with the possibility of generating additional income at reduced operating costs for DPCs (compared with the standard solutions for DPCs).
10. The opportunity to save on electrical energy and heating costs at existing coal mines.

However, in doing so, there are several risks and issues in implementation of the proposed CMM utilization options.

First, it should be noted that in each case, when considering CMM as a primary energy source, it is necessary to consider the variability of the CMM well rate and, if necessary, organize additional fuel reserves. This can lead to additional CAPEXs and a rise in the need for the creation of a special department, thus increasing the OPEX.

In addition, an important component of the implementation of the "Coal-Energy-Information" concept is the need to amend standards, technical regulations, and other regulatory documents.

Finally, there is one more difficulty, which is the global deficit of semiconductors and the related limitation of server hardware supplies. This problem can increase the deadlines for the delivery of DPCs and influence the architecture of server hardware.

Nevertheless, despite the mentioned difficulties, it is obvious that at present, there is a trend toward decarbonization in the world. However, not all countries imply the complete rejection of the use of coal. This means that for the coal mining industry, it is advisable to implement modern technologies and concepts that allow making coal mining as environmentally friendly as possible and allow diversifying the business of coal mining enterprises to ensure their stabile operation and financial sustainability.

Yet, over the coming 10–20 years, a significant decrease in coal production is not forecasted, which in turn determines some predictability of income for coal mining enterprises. It is logical to use part of this income for the implementation of different approaches for comprehensive development of coal deposits, assuming the simultaneous production of coal and extraction of CMM, which will allow achieving the goals of coal mining enterprises for environmental friendliness improvement and economic efficiency increases. In particular, the concept of "Coal-Energy-Information" can be considered one of such approaches.

## 7. Conclusions

Three options in the framework of the proposed concept (1 = DPC with CEC, 2 = DPC with AR, and 3 = DPC with TEC) which allow improving the reliability and efficiency of power supply services, decreasing the negative influence on the environment, and diversifying the business of coal mining enterprises were shown in the paper. All three options assumed construction of the DPC on the industrial site of a coal mining enterprise, with different solutions for the energy supply of this DPC and the technological equipment of the coal enterprise by all necessary types of energy (electricity, heat, and cooling), using CMM as the primary source of energy.

Each proposed option implies the creation of its own configuration of energy supply systems with the necessary power equipment, but implementation of any of the three options, on one hand, allows for reducing $CH_4$ emissions, contributing to respecting the environment, and on the other hand, makes it possible for coal mining enterprises to offer the market IT services as a new "product" and thus diversify production activities.

For each option, a primary feasibility study was conducted. CAPEX, OPEX, and profitability and savings calculations were made, and the payback periods of investments (five years for option 1, seven years for option 2, and six years for option 3) were determined. This paper also mentioned the major risks and issues inherent in the implementation of the proposed options, namely the necessity to consider the variability of the CMM wells rate, which determines the need to organize additional fuel reserves in some cases. Another potential difficulty is the global deficit of semiconductors and the related limitation of server hardware supplies.

Finally, it was pointed out that the construction of DPCs based on coal mines makes it possible to stimulate mining enterprises toward efficient CMM utilization and therefore improve the safety of mining activities, to increase economic efficiency, as well as to give impetus to the development of the IT industry in mining cities. In addition, it is noted that the proposed concept is in close correlation with the program for the development of the coal industry in Russia for the period up to 2035, since it allows one to create a base for the implementation of innovative technologies based on digital platforms that ensure the development of coal mining technology without the constant presence of personnel in underground mining workings, or "unmanned" coal mines.

**Author Contributions:** Methodology, F.N.; Formal analysis, A.S., K.V., F.N. and R.K.; Resources, A.S.; Writing—original draft, A.S., K.V., F.N. and R.K.; Writing—review & editing, S.C.; Visualization, F.N.; Project administration, K.V. All authors have read and agreed to the published version of the manuscript.

**Funding:** This research was supported by a state assignment of the Ministry of Science and Higher Education of the Russian Federation (No. 075-03-2021-138/3).

**Institutional Review Board Statement:** Not applicable.

**Informed Consent Statement:** Not applicable.

**Data Availability Statement:** Not applicable.

**Conflicts of Interest:** The authors declare no conflict of interest.

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
