# Peer review of "The Development of Coal Mine Methane Utilization Infrastructure within the Framework of the Concept “Coal-Energy-Information”"

_energies, doi:10.3390/en15238948_

Round 1
Reviewer 1 Report
The paper is fairly well written and may be published after some minor improvements. My requests for corrections are as follows:
Lines 16-17 and 23: please correct English
Lines 47-48: at least the authors should speculate about the effect of current costs on the profitability of their method (3-4 sentences on the coal and electricity prices in 2021 and 2022 would be OK to me)
Lines 75-76: Why is it like that? Explain please.
Line 106: How high is the emisssion? Please put a numer here.
Lines 132, 144, 151, 180-181: The citations are lacking
Line 177: 30m3/t? Is it tonne of a ‘raw’ or ‘dry’ material?
Line 188 (Caption of Fig. 2): The citation needed at the end of the sentence. It would also be good to show the daily production [t/day] of each of the mines indicated in Fig. 2.
Table 1: I'd suggest to replace ‘Solid partical size’ by ‘Particle size of solids’
Eq. (2): all the variables in the equation should be explained since the units of the right and left hand sides do not match.
Table 2: Explain and show the values you estimated or put a citation here. What are individual 'q' values - explain them since different subscripts are used compared to Eq.(3) for instance. Also please replace the word 'Gaz' with 'Gas'.
Line 292: The text should probably be Italic.
Line 304: There is lack of explanation what the authors mean by 'CDelta'
Table 3: what does 'The thermal correction' mean. Please explain it in the text.
Lines 316-321: start the numbers with '1'. The same comment refers to lines 343-351.
Lines 366-373: How do the data from Bulgaria affect the calculation results for Kuznieck region? You should make a 2-3 sentence explanation, at least. Or maybe it would be better to assume some example costs for a Russian plant?
Lines 442: 442: Are the costs assumed by the authors? If not then give a citation, please
Line 455: Please start the sentence with 'The'.
Reviewer 2 Report
1. There are many introductions about the research background and significance in the abstract of the paper. It is suggested to simplify and improve it, so as to highlight the innovative technology route or key achievements in the work of the paper.
2. The data in Figure 1 should be cited with its source or reference, as well as in Figure 3. In addition, whether the specific coal mine described in Figure 3 is a representative case should be explained.
3. The data in Table 1 are critical to the algorithm for choosing a method of utilization of CMM and evaluation of calculation results in the next step. It is recommended to supplement the sources of these threshold value such as sulfur and their scientificity and rationality.
4. Note that the number of gas composition in the text and formula should be subscript, such as CO2, H2O, etc.
Reviewer 3 Report
The manuscript presents objectives that are clearly stated and specified with an accurate description. The methodology has steps well selected. The work is clearly ambitious and goes beyond stata of the art, contributing also with novel concepts. The outputs of the paper improve in an efficient waythe expected impacts mentioned in the discussion of the state of the art. However the section 3 concerning to DPC construction in addition to the gas gensetsis moderate explained mainly in the part 3.3. Determination of DPC profitability that should be rwritten to clarify the achieved comparison
